# TD3: Tucker Decomposition Based Dataset Distillation Method for Sequential Recommendation

## ABSTRACT

In the era of data-centric AI, the focus of recommender systems has shifted from model-centric innovations to data-centric approaches. The success of modern AI models is built on large-scale datasets, but this also results in significant training costs. Dataset distillation has emerged as a key solution, condensing large datasets to accelerate model training while preserving model performance. However, condensing discrete and sequentially correlated user-item interactions, particularly with extensive item sets, presents considerable challenges. This paper introduces **TD3**, a novel **T**ucker **D**ecomposition based **D**ataset **D**istillation method within a meta-learning framework, designed for sequential recommendation. TD3 distills a fully expressive *synthetic sequence summary* from original data. To efficiently reduce computational complexity and extract refined latent patterns, Tucker decomposition decouples the summary into four factors: *synthetic user latent factor*, *temporal dynamics latent factor*, *shared item latent factor*, and a *relation core* that models their interconnections. Additionally, a surrogate objective in bi-level optimization is proposed to align feature spaces extracted from models trained on both original data and synthetic sequence summary beyond the naïve performance matching approach. In the *inner-loop*, an augmentation technique allows the learner to closely fit the synthetic summary, ensuring an accurate update of it in the *outer-loop*. To accelerate the optimization process and address long dependencies, RaT-BPTT is employed for bi-level optimization. Experiments and analyses on multiple public datasets have confirmed the superiority and cross-architecture generalizability of the proposed designs. Codes are released at https://anonymous.4open.science/r/TD3.

## CCS CONCEPTS

• **Information systems** → **Data mining**.

## KEYWORDS

Recommender Systems; Dataset Distillation; Bi-level Optimization

**ACM Reference Format:**
Anonymous Author(s). 2025. TD3: Tucker Decomposition Based Dataset Distillation Method for Sequential Recommendation. In *Proceedings of the ACM Web Conference 2025 (WWW '25), 28 April - 2 May, 2025, Sydney, Australia.* ACM, New York, NY, USA, 10 pages. https://doi.org/XXXXXXX.XXXXXXX

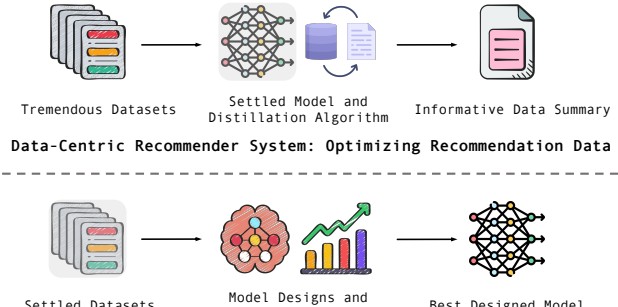

**Figure 1: Comparison of the data-centric recommender system through the lens of dataset distillation approach with the traditional model-centric recommender system. The key difference lies in their distinct optimization objectives.**

## 1 INTRODUCTION

To address the persistent challenge of information overload from the Internet, Sequential Recommendation Systems (SRS) capture users' evolving preferences through chronological interaction sequences [11, 17, 48, 55, 66, 74]. By recommending relevant items, these systems provide personalized services, alleviating users from having to seek out options themselves. However, as recommendation models become increasingly complex, the primary constraint affecting recommendation performance gradually shifts towards the quantity and quality of recommendation data [19], leading to the emergence of data-centric recommendations, as shown in fig. 1. Moreover, several emerging AI companies have prioritized data for its numerous benefits, including improved accuracy, faster deployment, and standardized workflows [34, 37, 46]. These collective initiatives in academia and industry highlight the growing recognition of data-centric approaches as essential for innovation.

Several initiatives have already been dedicated to the data-centric movement. A notable work launched by [65] aims to acquire an informative and generalizable training dataset for sequential recommender systems and asks the participants to iterate on the sequential recommendation dataset regeneration mostly focusing on improving the data quality. Another separate line is dataset distillation (DD) [49], which focuses on both data quality and data quantity. Unlike heuristic data pruning methods that directly select data points from original datasets, DD methods are designed to generate novel data points and have emerged as a solution for creating high-quality and informative data summaries. The utility of DD approaches has been witnessed in several fields, including federated learning [16, 26, 50, 59], continual learning [9, 32, 63], graph neural network [5, 10, 62, 69] and recommender systems [39, 40, 47].

Significant progress has been made in DD for non-sequential recommender systems [39, 47, 53, 54]. Methods like ∞-AE [39] and DConRec [53] distill user-item interaction matrices, while CGM [47] condenses categorical recommendation data in click-through rate (CTR) scenarios. Additionally, TF-DCon [54] employs large language models (LLMs) to condense user and item content for content-based recommendations. However, applying DD to sequential recommendation systems presents challenges due to several inherent complexities. (1) **Maintaining sequential correlations**: User-item interactions are sequentially correlated, reflecting the dynamic evolution of user preferences. Existing DD methods generate multiple synthetic interactions independently. Although it is possible to trivially organize these interactions into a sequence, this fails to capture the sequential correlations essential for modeling user behavior over time. (2) **Optimization dilemma**: In DD, the distilled dataset is typically parameterized as a learnable matrix, enabling fully differentiable distillation through a bi-level optimization process [2]. In sequential settings, this optimization becomes more difficult because the parameterized dataset size increases with sequence length. The enlarged parameter space further exacerbates convergence issues in the bi-level optimization process.

To address these challenges, we introduce TD3 to efficiently reduce computational complexity and extract streamlined latent patterns by decomposing the summary into four components: (1) *Synthetic User Latent Factor* (**U**), which represents synthetic user representations; (2) *Temporal Dynamics Latent Factor* (**T**), which captures temporal contextual information; (3) *Shared Item Latent Factor* (**V**), which characterizes items within the set and aligns with the item embedding table; and (4) *Relation Core* (**G**), which models the interrelationships among the factors. After decomposition, each factor is represented as a two-dimensional tensor, with its size determined by the sequence number, maximum sequence length, and item set size. Additionally, component **V** shared with the item embedding table does not require learning during distillation, making TD3 suitable for large item sets and long sequences. To address the final challenge, we propose an enhanced bi-level optimization objective to align feature spaces from models trained on both original and synthetic data. During *inner-loop* training, an augmentation technique allows the learner model to deeply fit the synthetic summary, ensuring accurate updates of it in the *outer-loop*. This approach accelerates convergence and, in conjunction with RaT-BPTT [6], minimizes computational costs while ensuring effective distillation. The contributions are concretely summarized:

- We study a novel problem in sequential recommendation: distilling a compressed yet informative synthetic sequence summary that retains essential information from the original dataset.
- We introduce TD3, which employs Tucker decomposition to separate the factors influencing the size of the synthetic summary, thereby reducing computational and storage complexity.
- Augmented learner training in *inner-loop* ensures precise synthetic data updates and feature space alignment loss is proposed beyond the naïve bi-level optimization objective for a better loss landscape to optimize while minimizing computational costs and preserving long dependencies through RaT-BPTT.
- Empirical studies on public datasets have confirmed the superiority and cross-architecture generalizability of TD3's designs.

## 2 RELATED WORK

**Sequential Recommendation (SR)** aims to leverage historical sequences to better capture current user intent [36]. In recent years, there has been rapid advancement in **model-centric** SR approaches, focusing from Markov chains [13] and factorization [38] to RNNs [14, 23], CNNs [43, 61], and GNNs [51, 56, 60]. Models like SASRec [17] and BERT4Rec [42] leverage self-attention mechanisms to learn the influence of each interaction on target behaviors. Recently, the emergence of LLMs has further enriched SR model design, for instance, [12, 64] leverage LLMs to uncover latent relationships, while RecFormer [22] represents items as item "sentences", and SAID [15] employs LLMs to learn semantically aligned item ID embeddings. With the emergence of the concept of **data-centric** recommendation, more works shift the focus to recommendation data enhancement. DR4SR [65] proposes to regenerate an informative and generalizable training dataset for sequential recommendations. FMLP-Rec [75] and HSD [68] adopt learnable filters for data denoising. DiffuASR [27] proposes a diffusion augmentation for higher quality data generation. ASReP [28] and MELT [18] focused on generating fabricated data for long-tailed sequences.

**Dataset Distillation (DD)** compresses large training datasets into smaller ones while preserving similar performance [8, 31]. There have been several lines of methods, that prioritize different aspects of information. **Performance matching** based methods focus on optimizing loss at the final training stage. For example, Farzi [40] distills auto-regressive data in latent space to produce a latent data summary and a decoder, although its parameters scale linearly with the vocabulary size and sequence length. ∞-AE [39] uses neural tangent kernels (NTKs) to approximate an infinitely wide autoencoder and synthesizes fake users through sampling-based reconstruction, while DConRec [53] distills synthetic datasets by sampling user-item pairs from a learnable probabilistic matrix, both tailored for collaborative filtering data in the form of user-item-rating triples. Another line of research focuses on **data matching**, encouraging synthetic data to replicate the behavior of target data. CGM [47], following the gradient matching paradigm [71] that mimics the influence on model parameters by matching the gradients of the target and synthetic data in each iteration, optimizes a new form of synthetic data rather than condensing discrete one- or multi-hot data in CTR scenarios. Furthermore, TF-DCon [54] utilizes large language models (LLMs) to condense item and user content for content-based recommendations, although this approach is hardly applicable to the contexts of ID-based sequential recommendation.

**Tucker Decomposition (TD)** decomposes a tensor into a set of factor matrices and a smaller core tensor [44]. It can be viewed as a kind of principal component analysis approach for high-order tensors. In particular, when the super-diagonal elements in the core tensor of tucker equal 1 and other elements equal 0, tucker decomposition degrades into canonical decomposition [58]. In three-mode case, A tucker decomposition of a tensor $X \in \mathbb{R}^{I_1 \times I_2 \times I_3}$ is:

$$X = \mathcal{G} \times_1 \mathbf{A}^{(1)} \times_2 \mathbf{A}^{(2)} \times_3 \mathbf{A}^{(3)} =: [\![\mathcal{G}; \mathbf{A}^{(1)}, \mathbf{A}^{(2)}, \mathbf{A}^{(3)}]\!], \quad (1)$$

where $\times_n$ indicating the tensor product along the n-th mode, each $A^{(n)} \in \mathbb{R}^{I_n \times R_n}$ is called the *factor matrix*, and $\mathcal{G} \in \mathbb{R}^{R_1 \times R_2 \times R_3}$ is the *core tensor*, show the level of interaction between all factors.

# 3 METHODOLOGY

This section introduces TD3, which distills discrete and complex sequential recommendation datasets into fully expressive synthetic sequence summaries in latent space.

## 3.1 Overview

**Notation.** Suppose we are given a large training dataset $\mathcal{T} \triangleq \{\mathbf{x}_i\}_{i=1}^{|\mathcal{T}|}$, where $\mathbf{x}_i \triangleq [x_{ij} \in \mathcal{V}]_{j=1}^{|\mathbf{x}_i|}$ is an ordered sequence of items, with each item $x_{ij}$ belonging to the set of all possible items $\mathcal{V}$. We denote the user set of the training dataset as $\mathcal{U}$, where $|\mathcal{U}| = |\mathcal{T}|$. Our goal is to learn a differentiable function $\Phi_\theta$ (*i.e.* SASRec) with parameters $\theta$, which predicts the next item $x_{i+1}$ given the previous sequence $x_{1:i}$. The parameters of this function are optimized by minimizing an empirical loss over the training set :

$$\theta^{\mathcal{T}} = \arg\min_\theta \ \mathcal{L}^{\mathcal{T}}(\theta) \, ,$$
$$\mathcal{L}^{\mathcal{T}}(\theta) \triangleq \mathbb{E}_{\mathbf{x} \sim \mathcal{T}, \, x_i \sim \mathbf{x}} \left[ \ell^{\mathcal{T}}(\Phi_\theta(\mathbf{x}_{1:i}), \ x_{i+1}) \right] \, , \tag{2}$$

where function $\ell^{\mathcal{T}}(\cdot, \cdot)$ represents the next item prediction loss, and $\theta^{\mathcal{T}}$ is the minimizer of $\mathcal{L}^{\mathcal{T}}$, which reflects the generalization performance of the model $\Phi_{\theta^{\mathcal{T}}}$. Our objective is to generate a small set of condensed synthetic sequence summary $\mathcal{S} \in \mathbb{R}^{\mu \times \zeta \times |\mathcal{V}|}$ consisting of $\mu$ fake sequences of maximum length $\zeta$ and $|\mathcal{S}| \ll |\mathcal{T}|$. Similar to eq. (2), once the condensed set is learned, the parameters $\theta$ can be trained on this set as follows :

$$\theta^{\mathcal{S}} = \arg\min_\theta \ \mathcal{L}^{\mathcal{S}}(\theta) \, ,$$
$$\mathcal{L}^{\mathcal{S}}(\theta) \triangleq \mathbb{E}_{\tilde{\mathbf{x}} \sim \mathcal{S}, \, \tilde{x}_i \sim \tilde{\mathbf{x}}} \left[ \ell^{\mathcal{S}}(\Phi_\theta(\psi(\tilde{\mathbf{x}}_{1:i}, \mathbf{E})), \ \tilde{x}_{i+1}) \right] \, , \tag{3}$$

where $\tilde{\mathbf{x}}_i \triangleq [ \, \mathcal{S}[i, j, :] \, ]_{j=1}^{\zeta}$, $\ell^{\mathcal{S}}(\cdot, \cdot)$ measures the distance between probability distributions, $\psi(\cdot, \cdot)$ is matrix product, $\mathbf{E}$ is the item embedding table and $\mathcal{L}^{\mathcal{S}}$ is the generalization performance of $\Phi_{\theta^{\mathcal{S}}}$. We wish the generalization performance of $\Phi_{\theta^{\mathcal{S}}}$ to be close to $\Phi_{\theta^{\mathcal{T}}}$ :

$$\mathbb{E}_{\mathbf{x} \sim \mathcal{T}, \, x_i \sim \mathbf{x}} \left[ \ell^{\mathcal{T}}(\Phi_\theta(\mathbf{x}_{1:i}), \ x_{i+1}) \right]$$
$$\simeq \ \mathbb{E}_{\tilde{\mathbf{x}} \sim \mathcal{S}, \, \tilde{x}_i \sim \tilde{\mathbf{x}}} \left[ \ell^{\mathcal{S}}(\Phi_\theta(\psi(\tilde{\mathbf{x}}_{1:i}, \mathbf{E})), \ \tilde{x}_{i+1}) \right] \, . \tag{4}$$

As the synthetic set $\mathcal{S}$ is significantly smaller, we expect the optimization in eq. (3) to be significantly faster than in eq. (2) .

**Problem.** The objective of achieving comparable generalization performance by training on synthetic data can be formulated in an alternative way. As proposed in [49], this can be framed as a meta-learning problem using bi-level optimization. In this approach, the *inner-loop* trains the learner models on synthetic data, while the *outer-loop* evaluates its quality using $\ell^{\mathcal{T}}(\cdot, \cdot)$ on the original dataset, updating the synthetic summary via gradient descent. More formally, the bi-level optimization problem can be expressed as :

$$\mathcal{S}^* = \mathbb{E}_{\theta_0 \sim \Theta} [\mathcal{L}^{\mathcal{T}}(\theta^*)] \, ,$$
$$\text{s.t.} \quad \theta^* = \arg\min_\theta \ \mathcal{L}^{\mathcal{S}}(\theta \mid \theta_0) \, . \tag{5}$$

The primary approach for addressing bi-level optimization problems is *truncated backpropagation through time* (T-BPTT) [35, 52] in reverse mode. When the *inner-loop* learner updated uses gradient

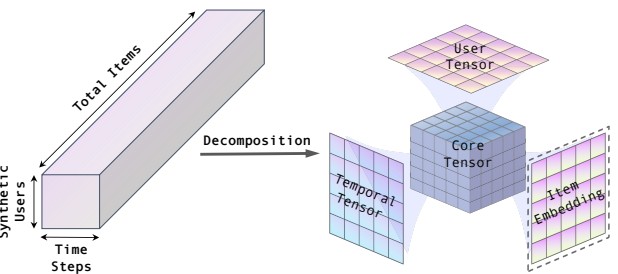

**Figure 2: An illustration of Tucker decomposition. The left part shows a three-dimensional synthetic sequence summary, with the third dimension representing the probability distribution over the entire item set. The right part illustrates the tucker decomposition, composed of a core tensor and factor matrices. The user tensor, temporal tensor, and core tensor are the parameters to be learned, while the item tensor shares values with the trained item embedding table.**

descent with a learning rate $\eta$, the meta-gradient with respect to the distilled sequence summary is obtained as follows :

$$\mathcal{G} = -\eta \frac{\partial \mathcal{L}^{\mathcal{S}}(\theta_T)}{\partial \theta} \sum_{i=T-M}^{T-1} \Pi_{j=i+1}^{T-1} \left[ 1 - \eta \frac{\partial^2 \mathcal{L}^{\mathcal{S}}(\theta_j)}{\partial \theta^2} \right] \frac{\partial^2 \mathcal{L}^{\mathcal{S}}(\theta_i)}{\partial \theta \partial \mathcal{S}} \, , \tag{6}$$

where $T$ represents the total optimization and unrolling steps that we perform in *inner-step* with loss $\mathcal{L}^{\mathcal{S}}(\theta)$, but T-BPTT only propagates backward through a smaller window of $M$ steps.

## 3.2 Synthetic Summary Decomposition

The discrete nature of user-item interaction records in sequential recommendation data complicates the direct use of gradient methods to distill an informative summary in the same format as the original data. Inspired by prior research [25, 30, 40], we choose to distill in the latent space. Consequently, we define the synthetic sequence summary as $\mathcal{S} \in \mathbb{R}^{\mu \times \zeta \times |\mathcal{V}|}$, a three-dimensional probability tensor that contains $\mu$ synthetic users, each with up to $\zeta$ interaction records. The third dimension of $\mathcal{S}$ represents the size of the entire item set $|\mathcal{V}|$, where $\mathcal{S}_{ij:}$ captures the interaction information of synthetic user $i$ at position $j$ by synthesizing the total original item information, with each item weighted differently, ensuring that the summary preserves the critical points.

Considering that the size of S is $\mu \times \zeta \times |\mathcal{V}|$, an increase in any of its dimensions leads to substantial growth in the overall tensor size, thereby escalating computational and storage requirements, especially when the original item set is large. Inspired by Tucker decomposition, $\mathcal{S}$ can be decomposed into the products of several smaller factor tensors and a core tensor, where the dimension of each sub-tensor is determined by only a single dimension, as visualized in fig. 2 and formalized as follows:

$$\mathcal{S} = G \times_1 \mathbf{U} \times_2 \mathbf{T} \times_3 \mathbf{V} =: [\![ G \, ; \, \mathbf{U}, \mathbf{T}, \mathbf{V} ]\!] \, , \tag{7}$$

where $\mathbf{U} \in \mathbb{R}^{\mu \times d_1}$, $\mathbf{T} \in \mathbb{R}^{\zeta \times d_2}$, $\mathbf{V} \in \mathbb{R}^{|\mathcal{V}| \times d_3}$, $G \in \mathbb{R}^{d_1 \times d_2 \times d_3}$, with $\times_n$ indicating the tensor product along the n-th mode and $d_n \ll |\mathcal{V}|$. Empirically, $d_1 = d_2$. More importantly, $\mathbf{V}$ is shared with the trained item embedding table, with no parameters needed to be trained.

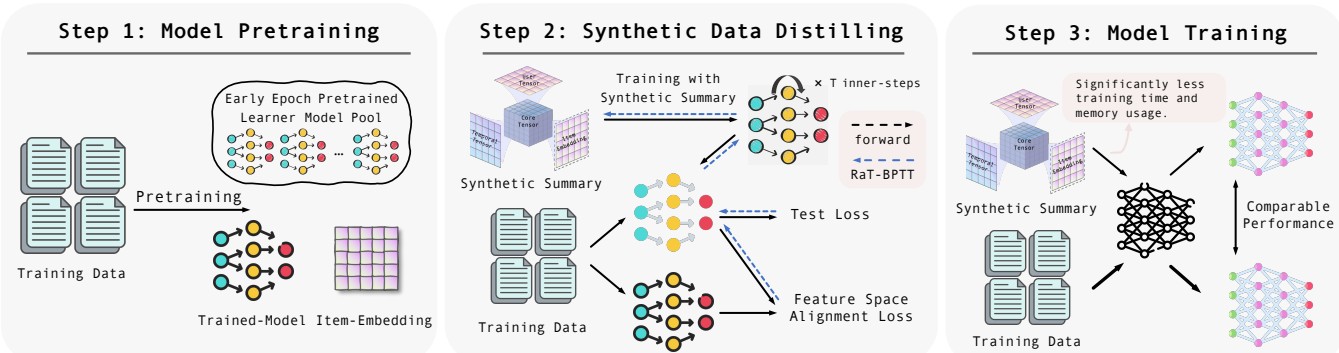

**Figure 3: Illustration of TD3. In step 1, the learner is trained to get the best checkpoint for feature space alignment and item embedding for the *shared item latent factor*, and checkpoints from early epochs are saved to form a pool for initialization in each *outer-loop*. Step 2 visualizes a single *outer-loop* step, where the meta-gradient is computed from both the test loss and the feature space alignment loss. Step 3 demonstrates that the distilled summary enables training of other networks with similar performance to models trained on real data, while significantly reducing training time and memory usage.**

The advantage of this decomposition lies in its ability to decouple the factors influencing the size of $\mathcal{S}$, thereby reducing data dimensionality as well as computational and storage complexity while preserving key feature information. This approach enables more efficient processing of high-dimensional data and enhances the understanding of its structure and characteristics.

## 3.3 Enhanced Bi-Level Optimization

Existing methods are computationally expensive for generating synthetic datasets with satisfactory generalizability, as optimizing synthetic data requires differently initialized networks [67]. To accelerate dataset distillation, we propose 1) *augmented learner training*, which enables the learner model to effectively fit the synthetic summary, supporting precise and comprehensive updates of synthetic data. Additionally, as mentioned in [33, 57], bias and poorly conditioned loss landscapes arise from truncated unrolling in TBPTT, we further propose 2) *feature space alignment* which aligns feature spaces from models trained on both original and synthetic data, combined with 3) *random truncated backpropagation through time* (RaT-BPTT) proposed by [6] to reduce bias in TBPTT and create a more favorable loss landscape for optimization.

*3.3.1 Augmented Learner Training.* As shown in eq. (7), we define the synthetic sequences summary as a three-dimensional probabilistic tensor obtained from the factors of the tucker decomposition and a core matrix via the mode product operation: $\mathcal{S} \in \mathbb{R}^{\mu \times \zeta \times |\mathcal{V}|}$. Under this settings, we use Kullback-Leibler (KL) Divergence as the loss function in eq. (3), and the inner objective is defined as:

$$\mathbf{x} = \mathcal{S}[\,:\,,\,:\,\zeta\,,\,:\,]\,,$$
$$\mathbf{y} = \mathcal{S}[\,:\,,\,\zeta\,,\,:\,]\,,$$
$$\hat{\mathbf{y}} = \text{Softmax}(\Phi_{\theta_j^{\mathcal{S}}}(\psi(\mathbf{x}, \mathbf{E})))\,, \qquad (8)$$

$$\ell^{\mathcal{S}}(\mathbf{y},\,\hat{\mathbf{y}}) = \mathcal{D}_{KL}(\mathbf{y} \,||\, \hat{\mathbf{y}}) = \sum_{i=1}^{|\mathcal{V}|} \mathbf{y}_i \log \frac{\mathbf{y}_i}{\hat{\mathbf{y}}_i}\,,$$

where $\mathbf{x}$ is the input, $\mathbf{y}$ is the target, $\Phi_{\theta_j^{\mathcal{S}}}(\cdot)$ is the learner model trained on synthetic data in $j$-th step, $\hat{\mathbf{y}}$ is output of the learner model and $\mathcal{D}_{KL}(\cdot \,||\, \cdot)$ is the discrete pointwise KL-divergence.

However, this approach only uses the previous 1 to $\zeta - 1$ interactions to predict the $\zeta$-th interaction probability. We propose further enhancing the prediction by randomly sampling middle positions, which can significantly improve the diversity of training, strengthen contextual understanding, enhance the model's generalization ability, and reduce reliance on specific sequence patterns. This strategy helps the model capture sequence information more comprehensively, improving its practical application performance.

*3.3.2 Feature Space Alignment.* We propose improving the current *outer-loop* test accuracy objective by incorporating a metric that ensures alignment between the feature spaces of models trained on both the original dataset and synthetic sequence summary. As noted in [20], the naïve meta-learning framework focuses primarily on matching the performance of models trained on original and synthetic data. However, from a loss surface perspective, this approach can be seen as mimicking the local minima of the target data using distilled data [21]. Nevertheless, it encounters considerable difficulties due to poorly conditioned loss landscapes [41]. Based on this defective, we aim to learn $\mathcal{S}$ such that the learner $\Phi_{\theta^{\mathcal{S}}}$ trained on them achieves not only comparable generalization performance to $\Phi_{\theta^{\mathcal{T}}}$ but also converges to a similar solution in the feature space. The enhanced objective can be formulated as:

$$\mathcal{S}^* = \arg\min_{\mathcal{S}} \ \mathbb{E}_{\theta_0 \sim \Theta} [\mathcal{L}^{\mathcal{T}}(\theta^*) + \mathcal{L}^{\mathcal{F}}(\theta^*)]\,,$$
$$\text{s.t.} \quad \mathcal{L}^{\mathcal{F}}(\mathcal{S}, \mathcal{T}; \theta^*) \triangleq \frac{1}{2} \| \Phi_{\theta^{\mathcal{T}}}(\mathcal{T}) - \Phi_{\theta^*}(\mathcal{T}) \|_F^2\,,$$
$$\mathcal{L}^{\mathcal{T}}(\theta) \triangleq \mathbb{E}_{\mathbf{x} \sim \mathcal{T},\, x_i \sim \mathbf{x}} \left[ \ell^{\mathcal{T}}(\Phi_{\theta^*}(\mathbf{x}_{1:i}),\, x_{i+1}) \right]\,, \qquad (9)$$
$$\theta^* = \arg\min_{\theta} \ \mathcal{L}^{\mathcal{S}}(\theta \mid \theta_0)\,,$$

where $\mathcal{L}^{\mathcal{F}}(\theta^*)$ is the feature space alignment loss with a mean squared error (MSE) by optimizing synthetic summary $\mathcal{S}$ directly. Unlike the most basic meta-learning framework, by aligning the feature spaces of models trained on both the original dataset and the synthetic summary, the synthetic summary achieves better generalization, going beyond simply matching performance metrics.

**Algorithm 1:** Optimization for TD3

**Input** : $\mathcal{T}$: original dataset; $[G, \mathbf{U}, \mathbf{T}, \mathbf{V}]$: core tensor and tucker factors for generating synthetic summary; $\Theta$: pretrained model parameters; $N$: total unrolling steps for BPTT; $\mathbf{W}$: truncated window size; $\alpha$: learning rate for the synthetic data; $\eta$: learning rate for the learner model;

1  // Outer loop: update synthetic sequences summary
2  **while** *not converged* **do**
3     ▷ Initialize learner's parameter $\theta_0 \sim \Theta$
4     ▷ Sample a mini-batch of original data $\mathcal{B}^{\mathcal{T}} \sim \mathcal{T}$
5     ▷ Uniformly sample the ending unrolling step $\mathbf{M} \sim U(\mathbf{W}, \mathbf{N})$
6     // Inner loop: update learner model parameters
7     **for** $n \leftarrow 1, \ldots, M$ **do**
8        ▷ Sample a mini-batch of synthetic user :
9           $\mathcal{B}^{\mathbf{U}} \sim \mathbf{U}$
10       ▷ Generate a mini-batch of synthetic summary :
11          $\mathcal{B}^{\mathcal{S}} = G \times_1 \mathcal{B}^{\mathbf{U}} \times_2 \mathbf{T} \times_3 \mathbf{V}$
12       // Start Random Truncked Backpropagation Through time
13       **if** $n = M - W - 1$ **then**
14          ▷ start accumulating gradients
15       **end if**
16       ▷ Update learner's parameter by gradient descent :
17          $\theta_n = \theta_{n-1} - \eta \, \nabla \mathcal{L}^{\mathcal{S}}(\mathcal{B}^{\mathcal{S}}; \theta_{n-1})$
18    **end for**
19    ▷ Compute test loss and feature space alignment loss :
20    $\mathcal{L}(\theta_M) = \mathcal{L}^{\mathcal{T}}(\theta_M) + \frac{1}{2} \parallel \Phi_{\theta^{\mathcal{T}}}(\mathcal{B}^{\mathcal{T}}) - \Phi_{\theta_M^{\mathcal{S}}}(\mathcal{B}^{\mathcal{T}}) \parallel_F^2$
21    ▷ Update synthetic data summary $\mathcal{S} = \mathcal{S} - \alpha \, \nabla_{\mathcal{S}} \mathcal{L}(\theta_M)$
22 **end while**
   **Output:** synthetic sequence summary $\mathcal{S}$.

*3.3.3 Random Truncated Backpropagation Through Time.* Training neural networks on distilled data is challenging largely due to the pronounced non-convexity of the optimization process. One common approach to capture long-term dependencies in this context is Backpropagation Through Time (BPTT), although it suffers from slow optimization and excessive memory demands. TBPTT, which limits unrolled steps, is a more efficient alternative. Yet, TBPTT introduces its drawbacks, such as bias from the truncation [57] and poorly conditioned loss landscapes, especially with long unrolls [45]. To address these issues, [6] propose the Random Truncated Backpropagation Through Time (RaT-BPTT) method, which combines randomization with truncation in BPTT. This approach unrolls within a randomly anchored and fixed-size window along the training trajectory and aggregates gradients within this window. The random window ensures that the RaT-BPTT gradient serves as a random subsample of the full BPTT gradient, covering the entire trajectory, while the truncated window improves gradient stability and reduces memory usage. As a result, RaT-BPTT enables faster training and better performance. It can be formulated as follows:

$$\mathcal{G} = -\eta \frac{\partial \mathcal{L}^{\mathcal{S}}(\theta_T)}{\partial \theta} \sum_{i=M-W}^{M-1} \Pi_{j=i+1}^{M-1} \left[ 1 - \eta \frac{\partial^2 \mathcal{L}^{\mathcal{S}}(\theta_j)}{\partial \theta^2} \right] \frac{\partial^2 \mathcal{L}^{\mathcal{S}}(\theta_i)}{\partial \theta \partial \mathcal{S}} , \quad (10)$$

where $M$ is the random number of total unrolled steps in the *inner-loop*, and $W$ represents the number of steps included in the backward, with only the final $W$ steps being used for backpropagation.

**Table 1: Statistical information of experimental datasets.**

| Dataset | # user ($|\mathcal{U}|$) | # item ($|\mathcal{V}|$) | # inter ($\sum_{\mathbf{x}} N_{\mathbf{x}}$) | avg. length | sparsity |
|---------|--------|--------|--------|--------|---------|
| Magazine | 408 | 758 | 2.7k | 6.6 | 99.13% |
| Epinions | 4739 | 7998 | 24.7k | 5.2 | 99.99% |
| ML-100k | 944 | 1683 | 100k | 106.0 | 93.71% |
| ML-1M | 6041 | 3707 | 1M | 165.6 | 95.53% |

Previous studies have demonstrated that diverse models in inner optimization improve robustness to overfitting [1, 70]. Moreover, pretrained models significantly enhance dataset distillation by providing better initialization, faster convergence, and higher-quality synthetic data [29, 40]. Building on these insights, we maintain a pool of pretrained learner models from early training epochs, capturing various stages of learning. At each outer step of the distillation process, we randomly sample from it to ensure a diverse and representative training signal for optimizing the synthetic dataset. The overall TD3 training procedure is summarized in Algorithm 1.

## 4 EXPERIMENTS

In this section, we present and analyze the experiments on four public datasets, aiming to answer the following research questions:

- **RQ1.** How does the performance of models trained on the synthetic summary, distilled using the TD3 method, compare to their performance when trained on the original data?
- **RQ2.** How does the performance of the synthetic data vary when applied to different recommendation model architectures that differ from the one used for dataset distillation?
- **RQ3.** What time savings are achieved by using the synthetic summary distilled by TD3 compared to training from scratch with the original sequential recommendation dataset?
- **RQ4.** How do the *Augmented Learner Training* module and *Feature Space Alignment* module in TD3 impact the efficiency and effectiveness of the entire dataset distillation process?

### 4.1 Settings

**Training Datasets.** To evaluate the distillation method proposed in this paper, we conduct experiments on four commonly used and publicly available datasets with variable statistics in table 1.

1) *Amazon*[1] includes Amazon product reviews and metadata infomation. For our empirical study, we mainly focus on the categories of "Magazine_Subscriptions".
2) *MovieLens*[2] is maintained by GroupLens and it contains movie ratings from the MovieLens recommendation service. We use the "ml-100k" and "ml-1m" versions for our experiments.
3) *Epinions*[3] was collected by [72] from Epinions.com, a popular online consumer review website. It describes consumer reviews and also contains trust relationships amongst users and spans more than a decade, from January 2001 to November 2013.

---

[1]http://snap.stanford.edu/data/amazon/productGraph/categoryFiles/
[2]https://grouplens.org/datasets/movielens/
[3]https://cseweb.ucsd.edu/~jmcauley/datasets.html#social_data

**Table 2: Comparison of TD3's performance (%) across various size synthetic data and the full dataset. Bold numbers denote the best-performing distilled summary. Underlined numbers denote that the distilled summary outperforms the full dataset. Superscript * means improvements are statistically significant with p<0.05, while ** meaning p<0.01.**

| Dataset & Model | Data size | HR@5 ↑ | HR@10 ↑ | HR@20 ↑ | NDCG@5 ↑ | NDCG@10 ↑ | NDCG@20 ↑ | MRR@5 ↑ | MRR@10 ↑ | MRR@20 ↑ |
|---|---|---|---|---|---|---|---|---|---|---|
| Magazine & SASRec | [10 x 10] ≡ 2.5% | 34.24 (±0.73) | 46.72 (±0.71) | 56.65 (±0.80) | 21.82 (±0.18) | 25.81 (±0.26) | 28.33 (±0.30) | 17.74 (±0.21) | 19.36 (±0.28) | 20.05 (±0.28) |
| | [15 x 15] ≡ 3.7% | **37.11** (±0.23)* | **48.60** (±0.90)* | 60.51 (±0.76)* | 24.14 (±0.21)** | 27.82 (±0.34)** | 30.82 (±0.35)** | 19.88 (±0.35)** | 21.38 (±0.34)** | 22.20 (±0.36)** |
| | [25 x 20] ≡ 6.1% | 34.15 (±1.21) | 47.45 (±0.31)* | 61.00 (±1.03) | 23.04 (±0.46)* | 27.36 (±0.35)** | 30.78 (±0.07) | 19.39 (±0.39)** | 21.19 (±0.44)** | 22.12 (±0.36)** |
| | [30 x 20] ≡ 7.4% | 34.81 (±0.47) | 47.87 (±1.54) | **61.17** (±1.83)* | 23.62 (±0.28)** | 27.90 (±0.88)* | **31.25** (±0.98)** | **19.96** (±0.53)** | **21.75** (±0.77)* | **22.67** (±0.80)* |
| | Full-Data | 34.15 (±0.65) | 45.40 (±0.76) | 56.98 (±0.23) | 21.92 (±0.18) | 25.63 (±0.28) | 28.57 (±0.14) | 17.90 (±0.12) | 19.47 (±0.22) | 20.29 (±0.20) |
| Epinions & SASRec | [10 x 50] ≡ 0.2% | 12.07 (±0.10) | 19.69 (±0.09)* | 30.51 (±0.19) | 8.15 (±0.07) | 10.59 (±0.10) | 13.31 (±0.09)* | 6.86 (±0.11) | 7.85 (±0.13) | 8.60 (±0.12) |
| | [15 x 30] ≡ 0.3% | **12.35** (±0.10) | **19.86** (±0.10)** | **31.06** (±0.19)* | **8.27** (±0.04) | **10.67** (±0.05)* | **13.49** (±0.08)** | **6.93** (±0.09) | **7.91** (±0.09) | **8.67** (±0.10) |
| | [20 x 20] ≡ 0.4% | 12.26 (±0.30) | 19.37 (±0.26) | 30.87 (±0.19)* | 8.20 (±0.19) | 10.47 (±0.20) | 13.38 (±0.12)* | 6.88 (±0.19) | 7.80 (±0.21) | 8.59 (±0.19) |
| | Full-Data | 11.83 (±0.42) | 19.13 (±0.16) | 30.09 (±0.27) | 7.92 (±0.29) | 10.25 (±0.19) | 13.00 (±0.08) | 6.64 (±0.25) | 7.58 (±0.21) | 8.33 (±0.17) |
| ML-100k & SASRec | [25 x 150] ≡ 2.6% | 48.57 (±0.23) | 66.45 (±1.27) | 81.30 (±0.28) | 32.64 (±0.30) | 38.44 (±0.69) | 42.20 (±0.35) | 27.39 (±0.42) | 29.80 (±0.56) | 30.83 (±0.47) |
| | [30 x 50] ≡ 3.2% | 49.84 (±1.02) | 66.14 (±1.16) | 81.37 (±0.44) | 33.30 (±0.31) | 38.56 (±0.86) | 42.44 (±0.72) | 27.88 (±0.63) | 30.04 (±0.85) | 31.12 (±0.82) |
| | [50 x 50] ≡ 5.3% | **51.86** (±0.13) | **68.79** (±0.49) | 82.96 (±0.36) | 35.13 (±0.59) | 40.62 (±0.16) | **44.21** (±0.20) | 29.63 (±0.38) | 31.90 (±0.19) | 32.89 (±0.22) |
| | Full-Data | 51.75 (±1.17) | 69.46 (±0.26) | 84.37 (±0.22) | 35.57 (±0.60) | 41.34 (±0.49) | 45.12 (±0.54) | 30.24 (±0.70) | 32.65 (±0.70) | 33.69 (±0.72) |
| ML-1m & SASRec | [50 x 50] ≡ 0.8% | 56.33 (±0.43) | 69.71 (±0.37) | 81.88 (±0.34) | 41.69 (±0.24) | 46.02 (±0.17) | 49.10 (±0.14) | 36.83 (±0.20) | 38.63 (±0.16) | 39.47 (±0.15) |
| | [100 x 50] ≡ 1.7% | 56.68 (±0.13) | 70.42 (±0.20) | 82.82 (±0.27) | 41.42 (±0.18) | 45.87 (±0.11) | 49.02 (±0.20) | 36.37 (±0.21) | 38.22 (±0.18) | 39.08 (±0.20) |
| | [200 x 100] ≡ 3.3% | **62.80** (±0.31) | **74.45** (±0.17) | **84.83** (±0.08) | **47.82** (±0.24) | **51.61** (±0.20) | **54.24** (±0.18) | **42.84** (±0.23) | **44.42** (±0.21) | **45.14** (±0.21) |
| | Full-Data | 67.92 (±0.36) | 78.22 (±0.28) | 86.91 (±0.09) | 54.02 (±0.14) | 57.37 (±0.11) | 59.57 (±0.07) | 49.38 (±0.09) | 50.78 (±0.07) | 51.38 (±0.07) |

**Evaluation Metrics.** We adopt the leave-one-out strategy for evaluation, following prior research [3, 65, 66]. For each sequence, the most recent interaction is used for testing, the second for validation, and the rest for training. To expedite evaluation, as in previous studies [17], we randomly sample 100 negative items to rank with the ground-truth item, which closely approximates full-ranking results while significantly reducing the computational cost. We assess performance using HR, NDCG, and MRR. HR@k checks if the target item appears within the top-k recommendations, NDCG@k considers the item's rank, and MRR@k computes the average reciprocal rank of the first relevant item, with $k \in \{5, 10, 20\}$.

**Implementation Details.** We implement TD3 using PyTorch and develop recommendation models based on the library of Recbole [73]. Throughout the distillation process, we use SASRec [17] serves as the learner across all datasets, utilizing attention heads $\in \{1, 2\}$, layers $\in \{1, 2\}$, hidden size $\in \{64, 128\}$, inner size $\in \{64, 128, 256\}$, and attention dropout probability of 0.5 and hidden dropout probability of 0.2. For magazine and epinions dataset, we set $d1, d2 \in \{8, 16\}$, while for ml-100k and ml-1m dataset, we set $d1, d2 \in \{16, 32, 64\}$. To evaluate the cross-architecture generalization of the proposed TD3, we employ GRU4Rec [14], BERT4Rec [42], and NARM [23] for performance assessment. For all distilled datasets, we apply the Adam optimizer [4] for both the *inner-loop* and *outer-loop* optimization.

In the *outer-loop*, the synthetic sequence summary optimizer is configured with a learning rate $\alpha \in \{0.01, 0.03\}$ and a weight decay of 0.0001, using a cosine scheduler to adjust the learning rate throughout the process. In the *inner-loop*, the learner optimizer employs a learning rate $\eta \in \{0.003, 0.005, 0.01\}$, with a weight decay of 0.00005. The inner steps are set to 200, using a random truncated window of 40 for backpropagation through time in RaT-BPTT, implemented via the Higher [7] package across all datasets.

## 4.2 Overall Performance

We evaluated TD3's performance across various synthetic sequence summary sizes and diverse datasets. In table 2, we compare models trained on full original datasets with those using various-sized synthetic sequence summaries. Additionally, table 3 and fig. 4 compare TD3's performance with Farzi and heuristic sampling methods: *random sampling*, which selects sequences uniformly, and *longest sampling*, which selects sequences in descending order of length. Our findings are as follows: 1) TD3 achieves comparable training performance, even with substantial data compression. This shows that small-batch synthetic summaries distilled from the original dataset effectively capture essential information, preserving data integrity for model training. 2) In datasets such as Magazine and Epinions, models trained on TD3-distilled summaries outperform those trained on the original datasets. This highlights the value of high-quality, smaller datasets over larger, noisier ones, underscoring the importance of data quality in model training. 3) Epinions, with the fewest average user-item interactions, achieved the highest data compression rate with TD3 while maintaining strong performance. This demonstrates data distillation's potential to address data sparsity in sequential recommendation tasks. 4) As illustrated in table 3 and fig. 4, TD3 is more sample-efficient than Farzi and heuristic methods, demonstrating superior data utilization.

These results illustrate the transformative potential of data distillation in improving sequential recommendation systems. This approach represents a shift towards a data-centric paradigm in recommender systems, where prioritizing data quantity and quality and strategic compression can create more robust and efficient algorithms, reducing computational costs and storage demands. This evolution paves the way for the next generation of recommendation algorithms, focusing on maximizing value from the minimal data.

**Table 3: Comparison of TD3 with existing dataset distillation techniques and heuristic sampling methods. "Random" indicates random sampling of the same number of interactions as used in dataset distillation. "Longest" refers to selecting the same number of longest user sequences.**

| Dataset | Metric | Sampling | | Distillation | | Full-Data |
|---|---|---|---|---|---|---|
| | | Random. | Longest. | Farzi | TD3 | |
| Magazine [30×20] | HR@10 ↑ | 15.44 (±2.68) | 19.62 (±0.31) | 41.46 (±1.27) | __47.87__ (±1.54) | 45.40 |
| | HR@20 ↑ | 27.50 (±1.03) | 33.66 (±2.73) | 58.70 (±0.13) | **61.17** (±1.83) | 56.98 |
| | NDCG@10 ↑ | 7.75 (±1.16) | 9.58 (±0.58) | 24.27 (±0.20) | __27.90__ (±0.88) | 25.63 |
| | NDCG@20 ↑ | 10.75 (±0.77) | 13.10 (±0.30) | 28.65 (±0.13) | **31.25** (±0.98) | 28.57 |
| Epinions [15×30] | HR@10 ↑ | 10.67 (±0.90) | 10.24 (±0.21) | 18.99 (±0.30) | __19.86__ (±0.10) | 19.13 |
| | HR@20 ↑ | 20.25 (±1.25) | 20.01 (±0.41) | 29.82 (±0.39) | **31.06** (±0.19) | 30.09 |
| | NDCG@10 ↑ | 4.93 (±0.36) | 4.79 (±0.06) | __10.38__ (±0.17) | __10.67__ (±0.05) | 10.25 |
| | NDCG@20 ↑ | 7.31 (±0.45) | 7.22 (±0.11) | __13.09__ (±0.07) | **13.49** (±0.08) | 13.00 |
| ML-100k [30×50] | HR@10 ↑ | 10.85 (±2.30) | 13.36 (±0.45) | 62.92 (±1.39) | **66.14** (±1.16) | 68.93 |
| | HR@20 ↑ | 21.49 (±3.98) | 25.59 (±0.64) | 77.84 (±0.30) | **81.37** (±0.43) | 83.78 |
| | NDCG@10 ↑ | 4.81 (±1.10) | 5.97 (±0.32) | 34.92 (±0.71) | **38.56** (±0.86) | 40.97 |
| | NDCG@20 ↑ | 7.48 (±1.28) | 9.02 (±0.36) | 38.72 (±0.40) | **42.44** (±0.72) | 44.76 |
| ML-1M [200×50] | HR@10 ↑ | 15.88 (±0.22) | 16.60 (±0.51) | 38.01 (±0.98) | **70.52** (±0.36) | 79.32 |
| | HR@20 ↑ | 28.09 (±0.31) | 30.93 (±0.45) | 56.10 (±1.24) | **82.52** (±0.25) | 87.60 |
| | NDCG@10 ↑ | 7.40 (±0.11) | 7.77 (±0.16) | 19.85 (±0.49) | **45.93** (±0.37) | 58.82 |
| | NDCG@20 ↑ | 10.47 (±0.10) | 11.36 (±0.14) | 24.41 (±0.55) | **48.97** (±0.30) | 60.93 |

**Table 4: Evaluation of generalization performance on unseen architectures using a synthetic summary of size $[50 \times 20]$ distilled from the Epinions dataset via SASRec.**

| Metric | Architecture | | | |
|---|---|---|---|---|
| | Synthetic Data / Original Data | | | |
| | SASRec | NARM | GRU4Rec | BERT4Rec |
| HR@10 | __19.75__ / 19.61 | __19.60__ / 19.43 | 19.11 / 20.25 | __18.98__ / 17.14 |
| HR@20 | __31.30__ / 30.90 | 30.55 / 31.10 | 29.49 / 31.49 | __29.08__ / 27.62 |
| NDCG@10 | __10.61__ / 10.45 | __10.67__ / 10.25 | 10.55 / 10.93 | __10.31__ / 9.29 |
| NDCG@20 | __13.51__ / 13.28 | __13.43__ / 13.18 | 13.15 / 13.75 | __12.83__ / 11.91 |
| MRR@10 | __7.85__ / 7.69 | __7.98__ / 7.48 | 7.96 / 8.13 | __7.70__ / 6.93 |
| MRR@20 | __8.64__ / 8.46 | __8.73__ / 8.28 | 8.66 / 8.90 | __8.38__ / 7.63 |

### 4.3 Cross-Architecture Generalization

Since the synthetic sequence summary is carefully tailored for optimizing a specific learner model, we assess its generalizability across various unseen architectures, as shown in table 4. We first distill the Epinions dataset using SASRec [17], resulting in a condensed synthetic summary of size $[50 \times 20]$. This summary is then used to train several alternative architectures, including GRU4Rec [14], which models sequential behavior using Gated Recurrent Units (GRU); NARM [24], which enhances GRU4Rec with an attention mechanism to emphasize user intent; and BERT4Rec [42], which uses bidirectional self-attention to learn sequence representations. The models trained on the synthetic summary demonstrate strong generalization across diverse architectures, maintaining high predictive performance. In some cases, they even outperform models trained on the original dataset, highlighting the effectiveness of our proposed TD3 method in enabling cross-architecture transferability.

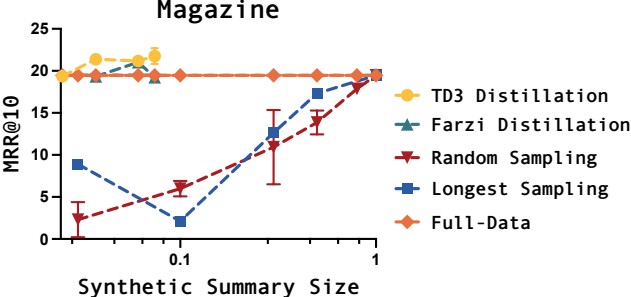

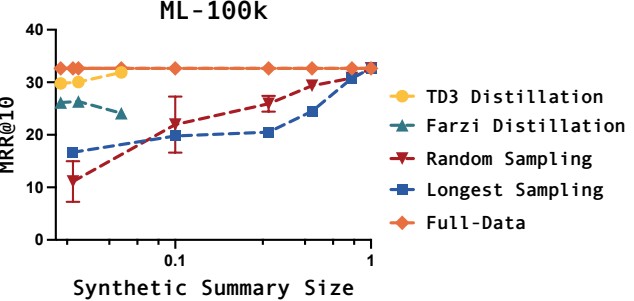

**Figure 4: Illustration of the performance comparison of the TD3 against: *Farzi, random sampling* and *longest sampling*. The above panel presents a comparison of the Magazine, while the below is on the ML-100k. For the *random sample* method, we conducted three independent trials to mitigate the impact of randomness on the experimental outcomes.**

### 4.4 Time and Memory Analysis

*4.4.1 Theoretical Memory Complexity.* As discussed in Section 2, Farzi [40] decomposes synthetic data $\mathcal{S} \in \mathbb{R}^{\mu \times \zeta \times |\mathcal{V}|}$ to a latent summary $\mathbf{D} \in \mathbb{R}^{\mu \times \zeta \times d}$ and a decoder $\mathbf{M} \in \mathbb{R}^{d \times |\mathcal{V}|}$, where $d \ll |\mathcal{V}|$. To highlight the advantages of employing Tucker decomposition for three-dimensional data, we perform a comparative analysis of our proposed approach with that of Farzi. In particular, we investigate the computational footprint associated with the bi-level optimization framework by evaluating memory usage during a single outer-loop step, providing insights into the efficiency gains achieved through our method as follows:

$$\text{Farzi}: O\left(|\Phi| + |\mathcal{B}^{\mathcal{T}}| \cdot \zeta \cdot d_3 + |\mathcal{B}^{\mathcal{S}}| \cdot \zeta \cdot |\mathcal{V}| + \mu \cdot \zeta \cdot d + d \cdot |\mathcal{V}|\right)$$

$$\text{TD3}: O\left(|\Phi| + |\mathcal{B}^{\mathcal{T}}| \cdot \zeta \cdot d_3 + |\mathcal{B}^{\mathcal{S}}| \cdot \zeta \cdot |\mathcal{V}| + (\mu + \zeta) \cdot d_1 + d_1^2 \cdot d_3\right)$$

where $|\mathcal{B}^{\mathcal{T}}|$ and $|\mathcal{B}^{\mathcal{S}}|$ denote the batch size of real and synthetic data, respectively, while $d_3$ represents the item embeddings' hidden dimension. Additionally, $d$, $d_1$, and $d_3$ are of the same order of magnitude and much smaller than $|\mathcal{V}|$. When the item set is very large, or the distilled sequence summary requires larger values of $\mu$ and $\zeta$, the inequality $(\mu + \zeta) \cdot d_1 + d_1^2 \cdot d_3 \ll \mu \cdot \zeta \cdot d + d \cdot |\mathcal{V}|$ holds, the method proposed in our work will offer a spatial advantage.

**Table 5: Wall-clock runtime (in A100 GPU hours) and storage costs for each operation. Distillation time represents the total runtime for 30 epochs, though actual dataset distillation requires less computation. Training on the original data uses an early stopping strategy while training on the synthetic data is fixed at 200 or 500 epochs.**

| Dataset | TD3 distillation process | | Training on original data | | Training on synthetic data | |
|---|---|---|---|---|---|---|
| | Compute | Memory | Compute | Memory | Compute | Memory |
| Magazine | 21m 28s | 742MB | 1m 13s | 666MB | 25s | 534MB |
| Epinions | 55m 46s | 2156MB | 1m 27s | 932MB | 18s | 624MB |
| ML-100k | 1h 58m 24s | 3460MB | 5m 47s | 1910MB | 14s | 604MB |
| ML-1m | 2h 26m 59s | 9832MB | 57m 31s | 3482MB | 22s | 924MB |

**Table 6: Ablation performance of a model trained on a [50 × 50] synthetic summary distilled from the ML-100k. ✗ indicates a module was not used, while ✓ indicates the opposite.**

| Dataset | FSA | ALT | NDCG@5 | NDCG@10 | MRR@5 | MRR@10 |
|---|---|---|---|---|---|---|
| ML-100k | ✗ | ✗ | 33.27 $_{(\pm 0.65)}$ | 38.80 $_{(\pm 0.51)}$ | 27.90 $_{(\pm 0.65)}$ | 30.19 $_{(\pm 0.59)}$ |
| | ✓ | ✗ | 34.27 $_{(\pm 0.57)}$ | 40.06 $_{(\pm 0.33)}$ | 29.03 $_{(\pm 0.61)}$ | 31.44 $_{(\pm 0.51)}$ |
| | ✗ | ✓ | 34.45 $_{(\pm 0.79)}$ | 39.75 $_{(\pm 0.27)}$ | 28.84 $_{(\pm 0.62)}$ | 31.03 $_{(\pm 0.39)}$ |
| | ✓ | ✓ | **35.13** $_{(\pm 0.59)}$ | **40.62** $_{(\pm 0.16)}$ | **29.63** $_{(\pm 0.38)}$ | **31.90** $_{(\pm 0.19)}$ |

*4.4.2 Empirical Computational Complexity.* The data distillation process consumes considerable wall-clock time and GPU memory, so we conducted a detailed quantitative analysis of these requirements for both distillation and model training on the original and distilled datasets, as shown in table 5. Wall-clock time is reported in single A100 (80GB) GPU hours. While distillation generally takes longer and uses more memory than training on the original data, its cost is often amortizable in real-world scenarios where multiple models must be trained on the same dataset. The amortization, based on the ratios in the table, varies by dataset and distillation scale. Notably, for large datasets, where training time is typically lengthy, training on significantly reduced distilled data can shorten this process by several orders of magnitude. This trade-off substantially decreases future training time, making distillation a one-time cost that yields long-term benefits for various downstream tasks, such as hyperparameter tuning and architecture exploration. Hence, the distillation process and its amortization will be well justified.

## 4.5 Ablation Studies

To further analyze the effects of different components in our method, we perform ablation studies on *ML-100k* as an example. Besides the RaT-BPTT, the ablation results of components about *Feature Space Alignment* (FSA) and *Augmented Learner Training* (ALT) are summarized in table 6. We show that FSA and ALT are complementary to each other. After integrating ALT, TD3's performance improved significantly. This improvement is due to ALT's ability to enhance the learner model's contextual understanding and reduce reliance on specific sequence patterns, enabling the model to capture sequence information more comprehensively. As a result, the synthetic data undergoes more accurate and comprehensive

updates during the *outer-loop*. Moreover, incorporating FSA alone also brings performance improvements in every metric. From the loss perspective, the basic bi-level optimization approach, which only uses the test loss as the objective function in the *outer-loop*, may encounter considerable challenges due to poorly conditioned loss landscapes, which is the main bottleneck affecting distillation performance. FSA strengthens the objective function, ensuring the model converges to a similar solution in the feature space while preserving the comparable model's performance when trained on the original and synthetic summary, thereby providing robust guarantees for optimal synthetic data updates. Therefore, using both components simultaneously, with each contributing in the *inner-loop* and *outer-loop*, maximizes the benefits of the distillation process.

## 5 CONCLUSION

In this paper, we propose TD3 which distills a large discrete sequential recommendation dataset into an informative synthetic summary, which is decomposed into four factors inspired by Tucker decomposition in latent space. TD3 offers several advantages, including the decoupling of factors that influence the size of $\mathcal{S}$, thereby reducing data dimensionality, computational costs, and storage complexity while preserving essential feature information. Additionally, we introduce an enhanced bi-level optimization approach featuring an augmented learner training strategy in the *inner-loop*, ensuring the learner deeply fits the summary and a feature-space alignment surrogate objective in the *outer-loop*, ensuring optimal learning of synthetic data parameters. Experiments and analyses confirm the effectiveness and necessity of the proposed designs.

Despite these advantages, our work has certain limitations, particularly the computational complexity of the bi-level optimization process, which remains demanding for larger models and extensive datasets. However, the cost is often amortizable in real-world scenarios where multiple models need training on the same dataset. This trade-off significantly reduces future training time, making distillation a one-time cost with long-term benefits for various downstream tasks. As for our future works, we intend to develop a more time-efficient dataset distillation method that can scale to larger datasets without compromising performance. Additionally, we also plan to leverage dataset distillation to facilitate cross-domain knowledge transfer, enabling distilling information from one domain to be effectively reused in others and enhancing the framework's versatility across various recommendation contexts.

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
