# OpenReview forum: "TD3: Tucker Decomposition Based Dataset Distillation Method for Sequential Recommendation"
_ACM.org/TheWebConf/2025/Conference — WWW 2025 Poster_

### Official Review · Reviewer_wJ4H · 2024-11-26

**Novelty:** 5
**Technical Quality:** 5

**Review:**

**Summary:**

This paper proposes TD3, a dataset distillation (DD) method tailored for sequential recommendation with Tucker Decomposition, which distills sequential recommendation datasets into fully expressive synthetic sequence summaries. Specifically, TD3 incorporates three crucial components: tucker decomposition, enhanced bi-level optimization, and random truncated backpropagation through time.

**Strengths:**

- **Motivation.** Data distillation in sequential recommendation is important and practical.
- **Experiments.** Experiments are conducted on four datasets, with varying data distillation ratios, to verify the effectiveness of the proposed method.
- **Writing.** The paper is well-written, with complete mathematical formulation and figure illustration. Moreover, the Algorithm 1 is detailed.

**Weaknesses:**

- **Backbone.** Currently, the effectiveness is only verified on a single backbone SASRec. To showcase the effectiveness of sequential recommendation in a broader view, it would be more beneficial to consider more backbones such as GRU4Rec [1] and Caser [2].

**Questions:**

**Questions:**

Why do we not continue to scale up the data distillation ratio since there is still a rising trend?

**Reviewer Confidence:**

2: The reviewer is willing to defend the evaluation, but it is likely that the reviewer did not understand parts of the paper

**Scope:**

4: The work is relevant to the Web and to the track, and is of broad interest to the community

---

### Official Review · Reviewer_e1xT · 2024-11-28

**Novelty:** 4
**Technical Quality:** 4

**Review:**

In this paper, the authors propose TD3, a Tucker Decomposition-based Dataset Distillation method for Sequential Recommendation, which compresses user-item interaction data into an informative synthetic summary. TD3 leverages Tucker decomposition to reduce dimensionality and computational complexity, representing data with four factors: user latent factor (U), temporal dynamics (T), shared item latent factor (V), and a core tensor (G).
The method employs enhanced bi-level optimization, incorporating augmented learner training and feature space alignment (FSA) to improve generalization. It also uses Random Truncated Backpropagation Through Time (RaT-BPTT) to handle sequential dependencies efficiently.
Empirical results show that TD3 achieves comparable or better performance than models trained on full datasets, with significant reductions in data size, training time, and memory usage, demonstrating its scalability and generalization across recommendation architectures.
While TD3 significantly reduces training costs for downstream tasks by generating compact synthetic datasets, the initial distillation process is computationally intensive, particularly for large datasets or complex models. However, the authors justify this cost as a “one-time expense” in real-world applications, where the distilled dataset can be reused across multiple training scenarios, such as hyperparameter tuning or architecture exploration. This amortization of the initial cost makes the approach practical and impactful for long-term use in scenarios requiring repeated model training. A more detailed analysis of this trade-off would further highlight the method’s scalability and efficiency.
The work is technically robust, combining Tucker decomposition with dataset distillation in a novel way. The methodology is well-detailed. The experiments are thorough, covering multiple datasets and models, with clear metrics and comparative baselines.
The paper is mostly clear, and figures and tables effectively support the text, making the methodology and results easier to understand. However, certain sections, particularly those discussing mathematical formulations, could benefit from more intuitive explanations for a broader audience and additional details to clarify complex concepts and their practical implications.
In summary
- Strengths: Technical rigor, comprehensive experiments, innovative use of Tucker decomposition, practical relevance in reducing computational demands.
- Weaknesses: Limited discussion of potential drawbacks (e.g., computational cost of the distillation process itself), and some technical sections could be more accessible.

**Questions:**

In a context where data is continuously updated, does the synthetic dataset creation process need to be repeated every time the original dataset changes? Could this aspect be explored further, potentially incorporating methods for incremental updates to the synthetic dataset?

**Reviewer Confidence:**

3: The reviewer is confident but not certain that the evaluation is correct

**Scope:**

4: The work is relevant to the Web and to the track, and is of broad interest to the community

---

### Official Review · Reviewer_FDEr · 2024-12-02

**Novelty:** 5
**Technical Quality:** 4

**Review:**

The paper introduces TD3, a novel Tucker Decomposition-based Dataset Distillation method within a meta-learning framework for sequential recommendation, which distills large datasets into a fully expressive synthetic sequence summary to accelerate training while preserving model performance, and demonstrates superior efficiency and generalizability across multiple public datasets.

Strength:
1. The paper investigates an interesting problem, which distill datastes into summaries to help with effiicient model training and reduced storage cost.
2. The paper presents an Tucker Decomposition based method which decouples the sequential data into user, item, time-specific factors along with their relational core.
3. The method provides a good trade-off between the data size and the effectiveness for model traning.

Drawbacks:
1. While the work aims to reduce data size, the datasets used in this research appear relatively small, which may limit the ability to convincingly demonstrate the effectiveness of the proposed method on larger, more complex datasets. Expanding the experimental scope to include larger datasets would provide stronger evidence for the claims made.
2. The paper compares the proposed method primarily with one existing dataset distillation approach but does not explore comparisons with other relevant methods in the experiments. To strengthen the experimental validation, it would be beneficial to include a broader range of comparison techniques, thereby offering a more comprehensive evaluation of the method's performance and advantages.

**Questions:**

1. The performance achieved after applying dataset distillation on the Magazine dataset exceeded that of training with the full dataset, which is an unusual observation. Could this be attributed to the relatively limited size of the Magazine dataset, potentially leading to overfitting with the full data?

2. Given the employment of Bi-Level Optimization in the dataset distillation procedure, did you encounter any challenges related to training stability during the experimental phase? If so, how were these issues addressed, and what impact did they have on the final outcomes?

**Reviewer Confidence:**

3: The reviewer is confident but not certain that the evaluation is correct

**Scope:**

3: The work is somewhat relevant to the Web and to the track, and is of narrow interest to a sub-community

---

### Official Review · Reviewer_aWTe · 2024-12-02

**Novelty:** 6
**Technical Quality:** 6

**Review:**

$\textbf{Summary}: $The paper introduces a novel dataset distillation framework designed for sequential recommendation systems. The method leverages Tucker decomposition and meta-learning to efficiently distill synthetic data sequences while maintaining the sequential correlation and essential information of user-item interactions.

$\textbf{Strength: }$
1. Distillation method could capture the semantics of the original datasets and consistently improve recommendation performance with less data.
2. Clear theory illustration and diagrams.
3. Distillation methods generalizes well on cross-models.

$\textbf{Weakness: }$
1. Distilling the model requires significant more time than training on the full dataset.

**Questions:**

1. As seen on Table. 5, pre-training the SASRec distillation model costs significant more time than training the full dataset from scratch. It raises two questions: a. how many runs are required to pretrain a SASRec distillation model (hyper-parameter tuning)? b. Can you give us more illustration of how much a good synthetic dataset would help down-stream task like you mentioned in 4.4.2?

2. Does RaT-BPTT have optimization problem like RNN?

**Reviewer Confidence:**

3: The reviewer is confident but not certain that the evaluation is correct

**Scope:**

4: The work is relevant to the Web and to the track, and is of broad interest to the community